# Estimating the Accuracy of Mandible Anatomical Models Manufactured Using Material Extrusion Methods

**DOI:** 10.3390/polym13142271

**Published:** 2021-07-11

**Authors:** Paweł Turek, Grzegorz Budzik

**Affiliations:** Faculty of Mechanical Engineering and Aeronautics, Rzeszów University of Technology, 35-959 Rzeszów, Poland; gbudzik@prz.edu.pl

**Keywords:** polymer models, optical systems, material extrusion methods, additive manufacturing, accuracy, mandible

## Abstract

The development of new solutions in craniofacial surgery brings the need to increase the accuracy of 3D printing models. The accuracy of the manufactured models is most often verified using optical coordinate measuring systems. However, so far, no decision has been taken regarding which type of system would allow for a reliable estimation of the geometrical accuracy of the anatomical models. Three types of optical measurement systems (Atos III Triple Scan, articulated arm (MCA-II) with a laser head (MMD × 100), and Benchtop CT160Xi) were used to verify the accuracy of 12 polymer anatomical models of the left side of the mandible. The models were manufactured using fused deposition modeling (FDM), melted and extruded modeling (MEM), and fused filament fabrication (FFF) techniques. The obtained results indicate that the Atos III Triple Scan allows for the most accurate estimation of errors in model manufacturing. Using the FDM technique obtained the best accuracy in models manufactured (0.008 ± 0.118 mm for ABS0-M30 and 0.016 ± 0.178 mm for PC-10 material). A very similar value of the standard deviation of PLA and PET material was observed (about 0.180 mm). The worst results were observed in the MEM technique (0.012 mm ± 0.308 mm). The knowledge regarding the precisely evaluated errors in manufactured models within the mandibular area will help in the controlled preparation of templates regarding the expected accuracy of surgical operations.

## 1. Introduction

Traditional modeling of machine elements and parts is carried out using computer-aided design (CAD) systems [1], which are also commonly used in the design process of industrial products using reverse engineering (RE) methods [2,3,4,5,6]. The designer’s concept becomes a reality due to creating a model using computer numerical control (CNC) techniques [7,8,9]. However, to minimize costs and increase the efficiency of the prototyping process and testing of new solutions, additive manufacturing (AM) [10,11,12,13,14] and hybrid methods, e.g., rapid tooling (RT), are used [15,16,17,18]. In the process of manufacturing final models, metallic materials are still the most often used. However, it has now been observed that due to the continuous improvement of mechanical and functional properties, polymeric materials are also used in processes including injection molding [19], machining [20,21], electrical discharge machining [22,23], and plastic working [24]. The demand for polymers materials results from their many advantages, including low density, high chemical resistance, easy forming of complex shapes and large sizes, good strength, and low production costs. Due to the improvement of mechanical and functional properties, polymeric materials are also used in additive manufacturing. In the process of manufacturing models using 3D printing methods, polymeric materials can take the form of solid [25,26], liquid [27,28], or semi-liquid [29,30,31]. The manufactured polymeric models are mainly used in the automotive [32], aviation [33,34], and medical industries [35,36,37].

In recent years, there has been a rapid increase in 3D printing techniques in manufacturing anatomical structures of the mandible in planning operations, including the reconstruction of continuity of the mandible geometry. Without a precise fixation of the mandible sections after the bone resection process, it may lead to breathing, speech, and swallowing problems. Thanks to the use of 3D printing methods, the procedures significantly improved. Manufactured models are most often used to pre-bend a reconstructive titanium plate before surgery or plan places where the resection process of a bone will be carried out. It is assumed that the accuracy of manufactured anatomical models in the mandible area should be in a range ±0.25 mm [38,39]. In the process of manufacturing surgical templates of the anatomical structures of the mandible, polyamide and acrylic materials were often used [40,41,42,43]. Due to the high cost of powder bed fusion, vat polymerization, and material jetting processes, new technologies were analyzed that would minimize the cost of surgery preparation. In recent years, there has been an increase in the use of thermoplastic polymer materials such as acrylonitrile butadiene styrene (ABS), polycarbonate (PC), polylactic acid (PLA), polyethylene terephthalate (PET), and polyetheretherketone (PEEK) in the process of planning surgical procedures within the mandibular area [44,45,46,47]. These are a group of thermoplastic materials used in the melted and extruded methods.

Along with modern manufacturing methods, more demands are placed on coordinate metrology, which concerns assessing the accuracy of manufacturing 3D models [48,49,50,51]. Most often, in the process of determining the accuracy of a geometry, coordinate measuring machines (CMM) [52,53,54] or articulated arm coordinate measuring machines (AACMM) [55] are used. However, in models with very complex geometry (such as anatomical structures), the measurement using tactile methods is very time-consuming or impossible to carry out. Therefore, in such cases, optical coordinate measuring systems such as structure light (SL) [56,57,58], laser scanner (LS) [59,60], or micro computed tomography (μCT) [61,62,63] are used. Accuracy tests of optical measurement systems are mainly carried out by VDI/VDE 2634, ASME B89.4.22, and VDI/VDE 2630 Blatt 1.3 standards [64,65,66]. In assessing the accuracy of optical coordinate measuring systems, traditional standards such as ball bar or flat plane are used. Currently, there are no present studies of new solutions of measures that would allow the preparation of measurement procedures to check the optical measurement errors in terms of a specific type of geometry (such as anatomical structures) and the material used. Moreover, no comparative research has been undertaken to select which of the optical systems allows to obtain the most reliable results assessing the mandible’s accuracy of 3D printing models. The knowledge regarding estimating manufacturing errors of melted and extruded methods can help with the controlled preparation of templates and surgical instruments regarding the accuracy expected during operations. It can also provide significant support in the procedures to restore the continuity of the mandible geometry.

## 2. Materials and Methods

To develop a methodology of estimating the optical systems’ accuracy, a template geometrically similar to the left side of the mandible was used. The template model was designed (Figure 1a) and manufactured from aluminum alloy AW-7075 on the DMU 100monoBLOCK. In the research process, three optical coordinate measuring systems were tested. In the process of reconstructing anatomical models of the 12 left sides of the mandible (Figure 1b), the Digital Imaging and Communications in Medicine (DICOM) data from the Siemens Somatom Sensation Open 40 scanner was used. A traditional “Head Routine” scanning protocol was used during the measurement process, intended for the craniofacial area. In the stage of the segmentation process, the threshold value was set above 200 HU. To visualize the 3D models of the left side of the mandible, the isosurface method was used. This method is based on the marching cube (MC) algorithm. The final models were saved to the STereoLitography (STL) format. The reconstructed body and angle of the left side of the mandible were used in the research because it is the most frequently affected by tumors. To this surface, they pre-bend a reconstructive titanium plate before the surgery.

The process of verifying the accuracy of the manufacture of the mandibular template was carried out on a coordinate measuring machine (CMM). As a result of the measurement of the model manufactured on the five-axis machining center, the template errors were estimated. The next step was to measure the template on three types of optical measurement systems: Atos III Triple Scan structure light (SL) system; articulated arm (AA) MCA-II with a laser scanner (LS) MMD × 100; and micro-tomography system (μCT) Benchtop CT160Xi. This process aimed to develop a methodology for estimating manufacturing errors of the mandible anatomical models using these systems. The Atos III Triple Scan system comprises a stand holding the measuring head, fitted with a projector and two cameras. The measuring system also includes a rotating table and a computer workstation for processing the measured data. Evaluating the Atos III Triple Scan system performance was carried out by the VDI/VDE 2634 standards requirements and represents three tests. During the probing error shape ($P_{s}$) measurement, the deviation between the measured diameter sphere and the calibrated value was determined using the least-squares fit method by Equation (1):(1)$$P_{s}=D_{measured}-D_{calibrated}$$

The sphere distance error (SD) determines the difference between the estimated and calibrated distance between the centers of the two spheres. The measured distance is derived from the measured values obtained from multiple area-based probing by Equation (2):(2)$$SD=L_{measured}-L_{calibrated}$$

The flatness measurement error (F) was made on the standard plate and is the range of the signed distances of the measurement point from the best-fit plane calculated according to the least-squares method. In the next stage, tests are performed on the system’s template measurement geometry (Figure 2a). In the first step, the outer part of the template was measured (Figure 2b); in the second step, the internal part was measured (Figure 2c). The Atos III Triple Scan measured each piece every 30 degrees (12 counting steps). The resolution of the data during the measurement was 0.050 mm, and the process was fully automated. Then, two measured geometries representing a cloud point were fitted in GOM Professional software using the best-fit algorithm to present the final template model. To assess the repeatability of the measurement process, it was repeated five times. The difference between the maximum (0.002 mm) and minimum (0.001 mm) value of the standard deviation was around 0.001 mm.

Measurements of the template model were also performed with the Metris MCA II articulated arm with a laser head MMD × 100. Firstly, the accuracy of the arm against the ASME B89.4.22 standard was checked. Under the procedure, three tests were performed to check the arm and one for the laser head. The effective diameter performance (EDP) test was carried out by probing nine points around specific areas of a mounted gage ball. The routine was completed three times, and the maximum absolute deviation from the certified value of the ball was recorded as the test result. The final deviation between the measured diameter sphere and the calibrated value was determined using the least-squares fit method by Equation (3):(3)$$EDP=D_{measured}-D_{calibrated}$$

In the single point articulation (SPA) test, the probe is placed within a conical socket. Individual points are measured from multiple approach angles with the maximum articulation of all of the principal joints. Each point measurement is analyzed as a range of deviations about the average value for the point locations by Equation (4):(4)SPA=Range/2

The volumetric length accuracy (VLA) test is the most appropriate test for determining machine accuracy and repeatability. It involves measuring a certified length standard many times in several locations and orientations and compares the resultant measurements to the actual length—Equation (5):(5)$$VLA=L_{measured}-L_{calibrated}$$

In the case of a laser scanner, an accuracy test is determined by scanning a plane from various directions. The result is the maximum standard deviation of the scan data to fitted plane features. The best-fit plane is calculated according to the least-squares method. The measurement procedure of the template used three measuring steps. The first step focused on measuring the outer part (Figure 3a) and the second on the internal part (Figure 3b). The third step was to measure the places on the external and internal surfaces that could not be measured in the first and second steps (Figure 3c). The resolution of the data during the measurement of the template model was 0.050 mm. Then, three measured geometries representing a cloud point were fitted in Focus Inspection software using the best-fit algorithm to present the final template model. The measurement process was repeated five times to assess the repeatability. The difference between the maximum (0.020 mm) and minimum (0.015 mm) value of the standard deviation was approximately 0.005 mm.

The Benchtop CT160Xi (Nikon) tomography used in the research is not calibrated and does not have a maximum permissible error (MPE). Before each measurement, one must check it separately. This scanning error process is mainly carried out on the ball-bar standard. The probing error of size (PS) is calculated as the difference between the measured diameter and the calibrated diameter by Equation (6):(6)$$PS=D_{measured}-D_{calibrated}$$

The length measurement (E) error were performed on reference standards (ball plates). During the procedure, the deviation between the measured length and the calibrated value was determined by Equation (7):(7)$$E=L_{measured}-L_{calibrated}$$

The template of the mandibular (Figure 4) also verified the Benchtop CT160Xi (Nikon) system. The structure of the iso-voxel during measurements was characterized by the size of the pixel 0.050 mm × 0.050 mm and the layer thickness 0.050 mm. The geometry of the model was reconstructed using ITK-Snap software. The segmentation process was carried out using the region growing method. The 3D model was visualized using the isosurface method. Measurement of the template was carried out five times while maintaining the repeatability conditions. The difference between the maximum (0.050 mm) and minimum (0.030 mm) value of the standard deviation was approximately 0.020 mm.

In the next step, the template and the 12 anatomical models of the left side of the mandible were manufactured using fused deposition modeling (FDM), melted and extruded modeling (MEM), and fused filament fabrication (FFF) techniques. During the manufacture of the models, comparable layer thickness was used (Table 1). Additionally, each model during the manufacturing process was oriented in the same way in the 3D printer space (Figure 5a). This procedure aimed to ensure that the side surface of the mandible models was as accurately manufactured as possible. This is because these surfaces are most often pre-bended surgical plates before the operation. Based on the development of measurement procedures, it was implemented in the perspective of the template and the 12 anatomical models of the left side of the mandible, manufactured using melted and extruded methods (Figure 5b,c).

The process of verifying the accuracy of manufacturing models was carried out in the Focus Inspection software. The fitting process of the nominal model obtained at the RE/CAD design stage and the reference model created at the measurement stage using the optical systems were carried out using the best-fit algorithm with an accuracy of 0.001 mm. Evaluation of the quality of manufacture geometry was carried out using:
Calculate the arithmetic mean (mean deviation) by Equation (8):
(8)$$\bar{y}=\frac{1}{n}\sum_{i=1}^{n}x_{i}$$
Sample standard deviation—sigma (σ) by Equation (9):
(9)$$\sigma =\frac{1}{\left(n-1\right)}{\sqrt{\sum_{i=1}^{n}{\left(x_{i}-\bar{x_{}}\right)}^{2}}}^{}$$
Calculate the skewness value by Equation (10):
(10)$$skewness=\frac{\sum_{i=1}^{n}{\left(x_{i}-\bar{x_{}}\right)}^{3}}{\sigma^{3}}$$
Calculate the kurtosis value by Equation (11):
(11)$$kurtosis=\frac{\sum_{i=1}^{n}{\left(x_{i}-\bar{x_{}}\right)}^{4}}{\sigma^{4}}$$
where *n*—the number of measurements, $x_{i}$—*i*-th measured value, $\bar{x}$—mean value.

## 3. Results

The obtained results of verification of the optical systems by the standards included in VDI/VDE 2634, ASME B89.4.22, and VDI/VDE 2630 Blatt 1.3 are presented in Table 2. The values do not exceed the allowable limit errors. The measurement was performed on a coordinate measuring machine to estimate the procedure errors of the optical systems. The obtained result precisely estimated the template’s manufacturing error (−0.002 mm ± 0.015 mm). Comparing the results obtained on the coordinate measuring machine can conclude that the procedure performed on Atos III Triple Scan is more accurate compared to the Metris MCA II articulated arm with a laser head MMD × 100 and Benchtop CT160Xi (Nikon) system. In the case of Atos III Triple Scan, the measurement result was 0.008 mm ± 0.035 mm. In the articulated arm with a laser head and Benchtop CT160Xi (Nikon) system, the results were comparable to about ± 0.100 mm (Table 3).

In the next step, the template model was manufactured using fused deposition modeling (FDM), melted and extruded modeling (MEM), and fused filament fabrication (FFF) techniques. The same measurement procedure was used in the template model manufactured using 100 DMU MonoBlock. The obtained results of the template are presented in Figure 6 and Table 3 and Table 4. The Atos III Triple Scan—(SL) obtained the most similar results to the coordinate measuring machine (CMM) results. This fact makes the Atos III Triple Scan the most precise optical system in the diagnostic process of geometry dimensional similar to the mandible, manufactured using melted and extruded methods (Table 3 and Figure 6). In the case of the Metris MCA II articulated arm with a laser head MMD × 100—(AA-LS) and Benchtop CT160Xi (Nikon) systems—(μCT), the standard deviation value is much higher than the Atos III Triple Scan. It is 0.1 mm for each thermoplastic material used (Table 3 and Figure 6). The obtained results are characterized by small and medium positive and negative skew (Table 4). In the kurtosis values, it can be analyzed that most data distributions are mainly leptokurtic (Table 4).

The final stage of the research was to estimate the accuracy of 12 anatomical models of the left side of the mandible, manufactured using five types of polymer materials: ABS-M30, PC-10, PLA, PET, and ABS-plus. The average results are presented in Table 5 and Table 6 and Figure 7, Figure 8 and Figure 9. The procedure used three optical systems to assess the accuracy of the manufacture of anatomical models. The same measurement procedure was used as in the case of the template model. In the process of assessing the normality of the data, the Shapiro-Wilk test was used. The evaluated values for all distributions (Figure 7, Figure 8 and Figure 9) are higher than the critical value; as a result, we did not reject the hypothesis of normal distribution. Accordingly, all obtained distributions were treated as usual.

Considering the results obtained from the Atos III Triple Scan system, they are similar to the measurements carried out on a template using a coordinate measuring machine. This fact makes the Atos III Triple Scan the most precise optical system in the diagnostic process of the mandible geometry. In each analyzed result obtained on Atos III Triple Scan, unimodal distribution was observed (Figure 8 and Figure 9). The models manufactured using the FDM technique (0.008 ± 0.118 mm for ABS-M30 and 0.016 ± 0.188 mm for PC-10 material) obtained the best accuracy. A very similar value of the standard deviation of PLA and PET material was observed (about 0.180 mm). The worst results were observed in the MEM technique (0.012 mm ± 0.308 mm) (Table 5 and Figure 7). The obtained results are characterized by small and medium positive (ABS-M30, PLA, and ABS-plus material) and negative skew (PC-10 and PET material) (Figure 8 and Figure 9 and Table 6). In the kurtosis values, it can be analyzed that most data distributions are mainly leptokurtic (Figure 8 and Figure 9 and Table 6).

In the case of a Metris MCA II articulated arm with a laser head MMD × 100, significantly overestimated noticed values of the standard deviation about the Atos III Triple Scan. The models manufactured using the ABS-M30 material (0.018 ± 0.184 mm) obtained the best accuracy. The worst was PC-10 and PLA material (Table 5 and Figure 7). The obtained results are characterized mainly by small and medium negative skew, except the models manufactured using the ABS-M30 material (Figure 8 and Figure 9 and Table 6). It can analyze that most data distributions are mainly leptokurtic (Figure 8 and Figure 9 and Table 6). In each analyzed result obtained on a Metris MCA II articulated arm with a laser head, MMD × 100 unimodal distribution was observed (Figure 8 and Figure 9).

In the Benchtop CT160Xi, it was noticed that models manufactured using the PLA material (−0.028 mm ± 0.210 mm) obtained the best accuracy results; the worst were obtained in the case of ABS-M30 material (Table 5 and Figure 7). The results are characterized by small and medium positive skew (Figure 8 and Figure 9 and Table 6). It can be observed that most data distributions are mainly leptokurtic, except the models manufactured using the ABS-M30 material (Figure 8 and Figure 9 and Table 6). In each analyzed result obtained on the Benchtop CT160Xi (Nikon), unimodal distribution was observed (Figure 8 and Figure 9).

## 4. Discussion

Designing and manufacturing an anatomical model or surgical template is not a simple task. It is especially true for the craniofacial area, which consists of bone tissues with very complex geometry. Therefore, it is necessary to know the accuracy of manufacturing models of anatomical structures. Currently, 3D printing methods are used in the process of manufacturing models of anatomical structures. In manufacturing, surgical templates or the anatomical structures of the mandible, polyamide, and acrylic materials were often used [40,41,42,43]. In recent years, there has been an increase in thermoplastic polymer materials such as ABS and PLA to plan surgical procedures within the mandibular area [44,45,46,47]. The accuracy of the manufactured anatomical models is most often verified using optical coordinate systems. It is assumed that the accuracy of these models should be approximately +/−0.25 mm [38,39]. However, the current research on the accuracy of manufacturing of the mandible using 3D printing methods does not include measuring errors [67,68,69,70,71]. Thanks to the design and implementation of a standard similar to the mandible geometry in this research, it was possible to estimate errors at the digitalization stage accurately and obtain the most reliable results assessing the accuracy of 3D printing models of the mandible manufactured using melted and extruded methods.

In determining errors in the optical measurement systems, the priority was checking the procedures by the VDI/VDE 2634, ASME B89.4.22, and VDI/VDE 2630 Blatt 1.3. Many items in the literature take the same steps [72,73]. The most time-consuming research was to carry out the procedures verifying the accuracy of the measuring arm and the laser head. It was necessary to carry out individual preparatory activities and demonstrate the arm and later the laser head. In the case of the Benchtop CT160Xi (Nikon) system, the procedure was the simplest. It was required to measure the geometry of the standard in the working space of the measuring system without additional preparation. In the Atos III Triple Scan system, the measuring table has significantly simplified the entire accuracy verification process. The obtained values for all methods were within recommended tolerances, confirming the correct functioning of the systems.

The first step estimated the procedure errors on a template geometrically similar to the mandible. It was performed on a coordinate measuring machine. The result very precisely estimated the manufacturing error of the template (−0.002 mm ± 0.015 mm). In the case of the optical systems, problems were observed. The surface of the template model manufactured from aluminum alloy was highly reflective. In the Atos III Triple Scan and Metris MCA II articulated arm with a laser head, it was decided to use matting spray from the titanium powder. Using this product yielded the thinnest layer (0.008 mm on average) and is proven to have the least influence on measurement accuracy in terms of dimension characteristics [74]. In the case of the Benchtop CT160Xi (Nikon), matting of the surface was not required. Still, due to the high radiological density of the aluminum alloy, the metal detection technique was used on the reconstruction stage of 2D images that allowed for the removal of metal artifacts [75]. Thanks to this process, better-quality images were obtained. Taking into account the obtained repeatability value, which consists of:
In the case of the Atos III Triple Scan: system errors, the adopted measurement procedure, and fitting errors of two-point cloudsIn the case of the Metris MCA II articulated arm with a laser head: system errors, the adopted measurement procedure and fitting errors of three-point cloudsIn the case of the Benchtop CT160Xi (Nikon): system errors, the adopted measurement procedure and the data processing stage (especially the segmentation and reconstruction process).

Atos III Triple Scan obtained the best results (the repeatability value was in the range of 0.001 mm) compared to the Metris MCA II articulated arm with a laser head (0.005 mm) and Benchtop CT160Xi (Nikon) (0.020 mm). Despite the obtained results, the Metris MCA II articulated arm with a laser head and Benchtop CT160Xi (Nikon) were included in assessing the template’s accuracy, and 12 models of the mandible were manufactured of thermoplastic materials. This procedure aimed to test all three optical systems for other types of materials.

At the stage of assessing the accuracy of the template model manufactured of thermoplastic materials, a CMM was also used. The results obtained on the CMM informed that the most accurate template model was manufactured on the Fortus 360-mc machine from ABS-M30 and PC-10 material. The FDM technique, high repeatability of the 3D printer, and materials used may have influenced the results. The template model manufactured using the MEM technique obtained the worst accuracy results. The FFF technique for PLA and PET materials obtained comparable results. It can be concluded that the results achieved on the Atos III Triple Scan presented in Table 3 and Figure 6 are the most similar to the CMM. In minimizing measurement errors, the automation of the head and the table played an important role. In two other optical systems, the results are not comparable to those obtained on the CMM and Atos III Triple Scan. Only in the template model manufactured from ABS-M30 material can we see a certain analogy. The presented results for the Metris MCA II articulated arm with a laser head and Benchtop CT160Xi (Nikon) are mainly influenced by measurement errors, which do not allow for a reliable assessment. Manual geometry measurement increases the value of the standard deviation in the case of a Metris MCA II articulated arm with a laser head MMD × 100. Additionally, to obtain a complete 3D geometry of the template, one more measurement was made than in the Atos III Triple Scan system. In the Benchtop CT160Xi (Nikon) system, a significant increase in the value of the standard deviation was also observed. Despite implementing one procedure for acquiring DICOM data, the accuracy reports differ significantly from the Atos III Triple system and the Metris MCA II articulated arm with a laser head MMD × 100. A different technique of geometry measurement mainly influenced the presented results compared to the Atos III Triple system and the Metris MCA II articulated arm with a laser head MMD × 100. Additionally, the final effect of the geometry reconstruction could have the data processing stage, especially the selection of threshold value at the segmentation stage [76,77,78,79].

In verifying the accuracy of the mandible geometry, no measurements were performed on the CMM. This is due to the complexity of the geometry and thus the time-consuming nature of the measure itself. An increase in statistical parameters observed in Table 5 and Figure 7 was comparable to the template model. This is natural because the increase in parameters results from the complexity of the geometry. However, between the results presented in Table 3 and Table 5, we can see an analogy. Mandible models manufactured using the FDM technique from ABS-M30 material obtained the best accuracy results. Considering the most accurately estimated results using the Atos III Triple Scan system, we can conclude that the recommended tolerance of +/−0.25 mm is met by the mandible models made of ABS-M30, PC, PET, and PLA material. In the case of the Metris MCA II articulated arm with a laser head and Benchtop CT160Xi (Nikon), it is difficult to reliably assess the accuracy of the models because measurement errors that were too large characterized both systems. Based on the presented research, the Atos III Triple Scan system should be recommended to verify the accuracy of the mandible geometry manufactured using melted and extruded methods.

## 5. Conclusions

Thanks to using a standard similar to the mandible, it was possible to precisely estimate the measurement errors of optical systems. This procedure allowed for selecting the system generating the smallest measurement errors occurring during the digitization of the mandible geometry. The research conducted in the article allowed for a reliable estimation of errors of anatomical models of the mandible manufactured using the melted and extruded methods.

The obtained results indicate that the Atos III Triple Scan allows for the most accurate estimation of errors in model manufacturing against a Metris MCA II articulated arm with a laser head MMD × 100 and Benchtop CT160Xi (Nikon) system. In minimizing measurement errors, the automation of the head and the table played an important role. It is possible to shorten the measuring time while reducing the number of measuring steps. Manual geometry measurement increases the standard deviation value in the case of a Metris MCA II articulated arm with a laser head MMD × 100. Additionally, to obtain a complete 3D geometry of the mandible, one more measurement was made than in the Atos III Triple Scan system. Considering the obtained value of standard deviation, one can make additional changes in the applied measurement methodology. However, due to the manual measurement process, it is difficult to maintain the high repeatability of the measurement. In the Benchtop CT160Xi (Nikon) system, a significant increase in the standard deviation value was observed. A different technique of geometry measurement mainly influenced the presented results compared to the other systems. In the case of tomography, we did not obtain a direct representation of the three-dimensional geometry of the digitized model. The final effect of the accuracy of geometry reconstruction could have the data processing stage, especially the selection of the threshold value at the segmentation stage.

Changing the measurement system affects the distribution of data on the histogram. Additionally, in the case of PC-10, PLA, PET, and ABS-plus materials, a significant change in the value of the skewness parameter was observed and the value of kurtosis also changed. However, leptokurtic distributions mainly dominated (this distribution is more peaked than a Gaussian distribution). Using the FDM technique (ABS-M30 and PC-10 material) obtained the best accuracy. A very similar value of the standard deviation of PLA and PET material was observed in the FFF technique. The worst accuracy results were observed in the case of the MEM technique. Considering the most accurately estimated results using the Atos III Triple Scan system, we can conclude that the recommended tolerance of +/−0.25 mm is met by the mandible models made of ABS-M30, PC, PET, and PLA material.

The knowledge regarding the procedures for estimating manufacturing errors can help in the future in controlled preparation of templates and surgical instruments in terms of the accuracy expected during operations. It can also provide significant support in the procedures to restore the continuity of the mandible geometry. The results are the starting point for further analyses related to developing measurement protocols that minimize digitalization error with other human anatomical area models.

## Figures and Tables

**Figure 1 polymers-13-02271-f001:**
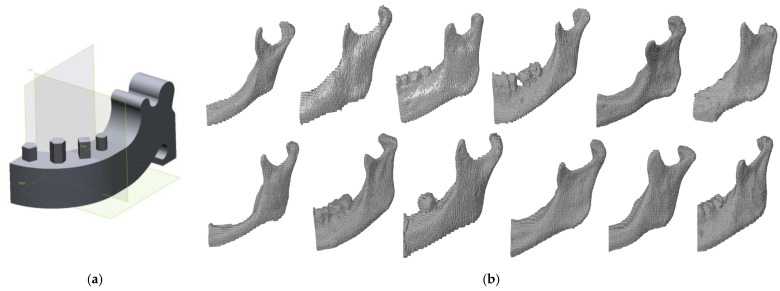
The final models representing: (**a**) template; (**b**) anatomical models of the left side of the mandible.

**Figure 2 polymers-13-02271-f002:**
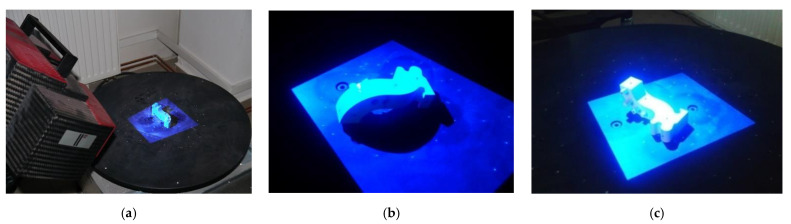
The measurement procedure performed on the Atos III Triple Scan system: (**a**) the measurement system; (**b**) measurement of the outer part; (**c**) measurement of the internal part.

**Figure 3 polymers-13-02271-f003:**
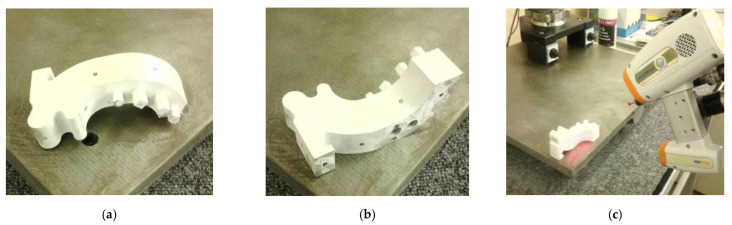
The measurement procedure performed on the articulated arm (MCA-II) with laser head (MMD × 100) system: (**a**) measurement of the outer part; (**b**) measurement of the internal part; (**c**) measurement of the outer and internal surface, which could not be measured in the first and second step.

**Figure 4 polymers-13-02271-f004:**
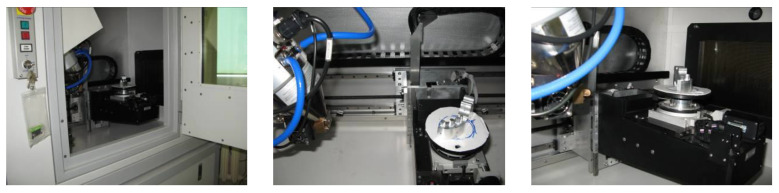
The measurement procedure performed on the Benchtop CT160Xi (Nikon) system.

**Figure 5 polymers-13-02271-f005:**
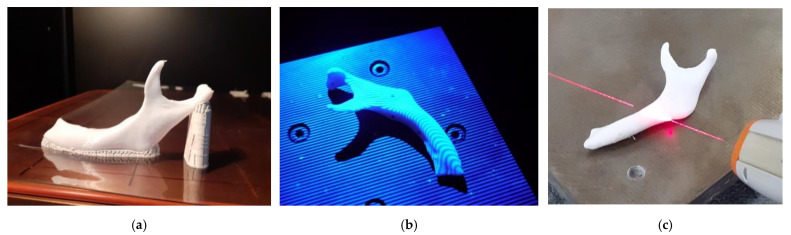
The example of manufacturing and measuring process of the polymer anatomical model: (**a**) orientation of a model in the 3D printer space; (**b**) the measurement procedure performed on the Atos III Triple Scan; (**c**) the measurement procedure performed on the articulated arm (MCA-II) with laser head (MMD × 100) system.

**Figure 6 polymers-13-02271-f006:**
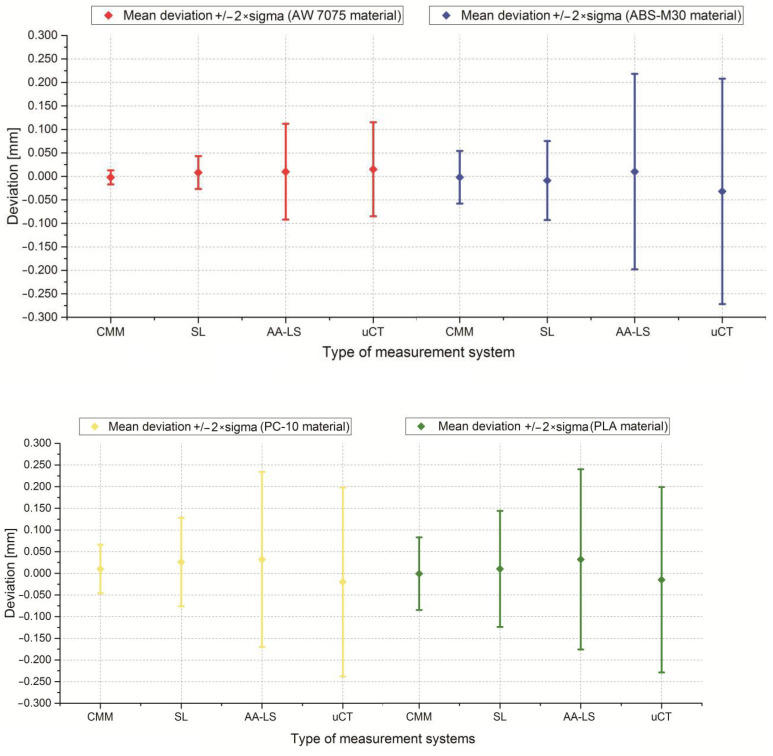
Statistical parameters representing the value of mean deviation +/− 2 × sigma of the template model.

**Figure 7 polymers-13-02271-f007:**
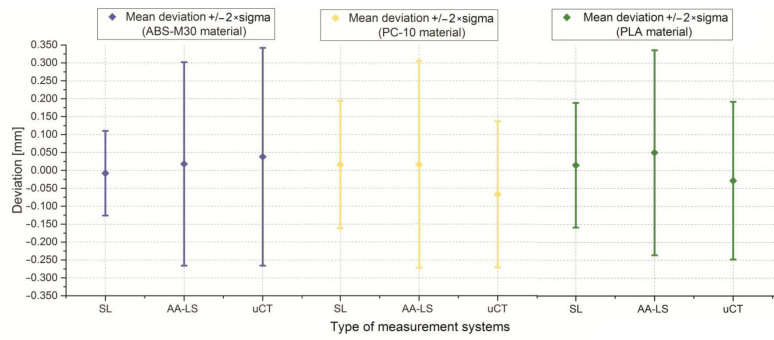
Statistical parameters representing the value of mean deviation +/− 2 × sigma of the 12 parts of the mandible model.

**Figure 8 polymers-13-02271-f008:**
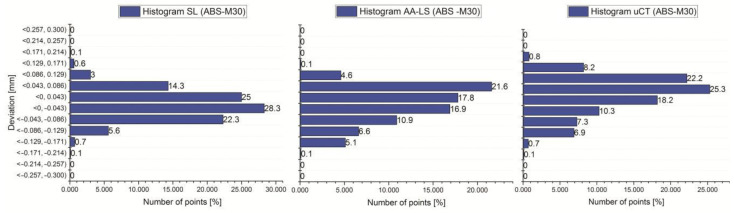
Histograms representing average results of the accuracy of the 12 parts of the mandible manufactured using ABS-M30, PC-10, PLA, and PET material.

**Figure 9 polymers-13-02271-f009:**
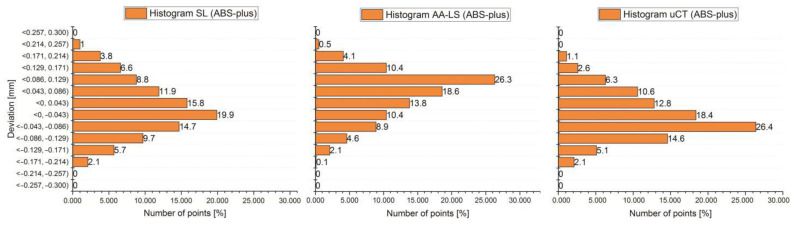
Histograms representing average results of the accuracy of the 12 parts of the mandible manufactured using ABS-plus material.

**Table 1 polymers-13-02271-t001:** AM technologies used in the research.

AM Technology	3D Printer	Commercial Material Name	Generic Name	Layer Thickness
Fused Deposition Modeling (FDM)	Fortus 360-mc	ABS-M30	Acrylonitrile	0.178 mm
			Butadiene	
			Styrene	
		PC-10	Polycarbonate	0.178 mm
Fused Filament Fabrication (FFF)	Prusa MK3s	PLA	Polylactic acid	0.150 mm
		PET	Polyethylene	0.150 mm
			terephthalate	
Melted and Extruded Modeling (MEM)	UP Box	ABS plus	Acrylonitrile	0.150 mm
			Butadiene	
			Styrene	

**Table 2 polymers-13-02271-t002:** The results of verification optical systems using the standard procedure.

Atos III Triple Scan	
Acceptance test according to VDI/VDE 2634	Measured value/Maximum permission error (2σ)
Probing error	±0.003 mm/±0.006 mm
Sphere—spacing error	±0.007 mm/±0.020 mm
	Maximum error (2σ)
Flatness measurement error	±0.020 mm
Articulated arm coordinate measuring machine (MCA II) with a laser head (MMD × 100)	
Acceptance test according to ASME B89.4.22	Measured value/Maximum permission error (2σ)
Effective diameter test	±0.004 mm/±0.008 mm
Single point articulation test	±0.022 mm/±0.024 mm
Volumetric performance test	±0.032 mm/±0.035 mm
Laser head test (flat plate)	±0.020 mm
Benchtop (Nikon) CT160Xi	
Acceptance test according to VDI/VDE 2630 Blatt 1.3	Measured value
Length measuring error	±0.006 mm
Probing error of size (Scanning error)	±0.006 mm

**Table 3 polymers-13-02271-t003:** Statistical parameters representing values of mean deviation +/− 2 × sigma of the template model.

Material	Coordinate Measuring Machine	Atos III Triple Scan	MCA II with Laser Head MMD × 100	Benchtop (Nikon) CT160Xi
AW 7075	−0.002 mm ± 0.015 mm	−0.008 mm ± 0.035 mm	0.010 mm ± 0.102 mm	0.012 mm ± 0.100 mm
ABS-M30	−0.002 mm ± 0.056 mm	−0.009 mm ± 0.084 mm	0.010 mm ± 0.208 mm	−0.032 mm ± 0.240 mm
PC-10	0.010 mm ± 0.056 mm	0.026 mm ± 0.102 mm	0.032 mm ± 0.220 mm	−0.020 mm ± 0.218 mm
PLA	−0.001 mm ± 0.084 mm	0.010 mm ± 0.134 mm	0.032 mm ± 0.210 mm	−0.015 mm ± 0.214 mm
PET	−0.010 mm ± 0.102 mm	0.008 mm ± 0.144 mm	0.028 mm ± 0.216 mm	−0.012 mm ± 0.250 mm
ABS-plus	−0.005 mm ± 0.212 mm	0.017 mm ± 0.270 mm	0.025 mm ± 0.204 mm	−0.008 mm ± 0.236 mm

**Table 4 polymers-13-02271-t004:** Statistical parameters representing values of skewness and kurtosis of the template model.

	Coordinate Measuring Machine					Atos III Triple Scan				
Material	ABS-M30	PC-10	PLA	PET	ABS-plus	ABS-M30	PC-10	PLA	PET	ABS-plus
Skewness	0.202	−0.252	0.152	−0.336	0.342	0.416	−0.694	0.342	−0.235	−0.235
Kurtosis	2.900	2.997	2.844	3.154	3.349	3.715	3.156	3.508	3.432	3.432
	MCA II with laser head MMD × 100					Benchtop (Nikon) CT160Xi				
Material	ABS-M30	PC-10	PLA	PET	ABS-plus	ABS-M30	PC-10	PLA	PET	ABS-plus
Skewness	0.235	−0.250	−0.289	−0.329	−0.326	0.532	0.345	0.225	0.463	0.254
Kurtosis	3.023	3.424	3.201	3.326	3.230	1.980	2.967	3.324	3.043	3.245

**Table 5 polymers-13-02271-t005:** Statistical parameters representing values of mean deviation ±2 × sigma of the 12 parts of the mandible model.

Material	Atos III Triple Scan	MCA II with Laser Head MMD × 100	Benchtop (Nikon) CT160Xi
ABS-M30	−0.008 mm ± 0.118 mm	0.018 mm ± 0.184 mm	0.038 mm ± 0.304 mm
PC-10	0.016 mm ± 0.188 mm	0.016 mm ± 0.290 mm	−0.067 mm ± 0.204 mm
PLA	0.015 mm ± 0.174 mm	0.050 mm ± 0.286 mm	−0.028 mm ± 0.210 mm
PET	0.012 mm ± 0.182 mm	0.056 mm ± 0.240 mm	−0.040 mm ± 0.264 mm
ABS-plus	0.012 mm ± 0.308 mm	0.040 mm ± 0.246 mm	−0.019 mm ± 0.260 mm

**Table 6 polymers-13-02271-t006:** Statistical parameters representing average values of skewness and kurtosis of the 12 parts of the mandible models.

	Atos III Triple Scan					MCA II with Laser Head MMD × 100				
Material	ABS-M30	PC-10	PLA	PET	ABS-plus	ABS-M30	PC-10	PLA	PET	ABS-plus
Skewness	0.201	−0.405	0.150	−0.402	0.323	0.391	−0.405	−0.321	−0.453	−0.487
Kurtosis	3.432	3.125	4.089	3.772	2.585	2.853	3.125	3.437	3.743	3.315
	Benchtop (Nikon) CT160Xi									
Material				ABS-M30	PC-10	PLA	PET	ABS-plus		
Skewness				0.421	0.610	0.402	0.501	0.327		
Kurtosis				2.341	3.921	3.502	3.324	3.497		

## Data Availability

The data presented in this study are available on request from the corresponding author.

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
