# Peer review of "Estimating the Accuracy of Mandible Anatomical Models Manufactured Using Material Extrusion Methods"

_polymers, 2021, doi:10.3390/polym13142271_

Round 1

Reviewer 1 Report

The authors present a manuscript of the methods for Estimating the Accuracy of Mandible Anatomical Models, which might be of interest to the readers of the journal. However, the following issues need to be addressed prior to acceptance for publication

  • The intro still needs a clear definition of what the objectives of the article will be.
  • Figure 2 is still too difficult to understand for the general reader of the journal.
  • I suggest removing the bullets from the conclusions section as they are not usually part of the style of the journal
  • There are still minor issues with the command of the English language.

Author Response

Reviewer 1

Thank you for your essential comments on the manuscript. Below are the answers to the questions.

  • The intro still needs a clear definition of what the objectives of the article will be.

The authors stated that it would be worth changing the title of the manuscript. The article's primary purpose was to precisely determine the accuracy of the models made using Melted and Extruded techniques. Additionally, the last few sentences in the introduction were edited to clarify the subject (red font color).

  • Figure 2 is still too difficult to understand for the general reader of the journal.

Figure 2 improved by presenting the visualization of the measuring system (Fig.2a).

  • I suggest removing the bullets from the conclusions section as they are not usually part of the style of the journal

The changes made to the conclusion section (red font color).

  • There are still minor issues with the command of the English language.

The authors made additional language improvements. If necessary, the authors ask for correct the text by the MDPI service.

Reviewer 2 Report

The manuscript 'A Methodology for Estimating the Accuracy of Mandible Anatomical Models Manufactured Using Material Extrusion Methods' cannot be published in the Polymers journal.

The polymers journal must maintain a high standing on the scientific articles that are published.

The manuscript submitted by the authors requires a lot of rework and should be rejected for publication in the Polymers Journal.

Some observations of my conclusion are:

The goal of the document is missing in the introduction section. It is possible to visualize what the authors intend, but the main goal must be clear and specify the research's objective.

The Figures presented throughout the document do not represent, in any case, a measurement procedure. The authors have to add the results obtained directly from the different systems used to measure and discuss them, rather than put just photos of the specimens, which occurs more often in undergraduate student Thesis.

I believe that this document requires more scientific data and a more in-depth discussion.

The document is entirely statistical, without presenting clear evidence of scientific analysis or technological development.

I recommend to the authors try to publish this document in another journal with a low impact factor and perhaps Q4 or in a conference, but it cannot be published in the Polymers Journal.

The document lacks scientific language, much less containing technical data or some interesting consideration to publish in this journal. Therefore, this document cannot be published in Polymers Journal or any other from MDPI.

Author Response

Reviewer 2

Thank you for your essential comments on the manuscript. Below are the answers to the questions.

  • The goal of the document is missing in the introduction section. It is possible to visualize what the authors intend, but the main goal must be clear and specify the research's objective.
  • The Figures presented throughout the document do not represent, in any case, a measurement procedure. The authors have to add the results obtained directly from the different systems used to measure and discuss them, rather than put just photos of the specimens, which occurs more often in undergraduate student Thesis.

The authors stated that it would be worth changing the title of the publication. The article's primary purpose was to precisely determine the accuracy of the models made using Melted and Extruded techniques. The knowledge regarding estimating manufacturing errors of Melted and Extruded methods can help the future in the controlled preparation of templates and surgical instruments regarding the accuracy expected during operations. It can also provide significant support in the procedures to restore the continuity of the mandible geometry.

  • I believe that this document requires more scientific data and a more in-depth discussion.
  • The document is entirely statistical, without presenting clear evidence of scientific analysis or technological development.

Currently, in the literature, two parameters are most often considered in the analysis of manufacturing accuracy, i.e., the mean deviation and the standard deviation. Therefore, the authors mainly focused on these parameters. Additionally, we conducted a test for the normality of the distribution, which will allow us to determine the expanded uncertainty (2sigma). Thanks to this, we have presented the accuracy of the models very precisely.

  • The manuscript submitted by the authors requires a lot of rework and should be rejected for publication in the Polymers Journal.

Due to one positive and one negative review, the journal's editor will appoint a third reviewer who will decide whether to accept or reject the manuscript.

Reviewer 3 Report

This paper addresses a broad audience, as the topic of medical prostheses for human bones is of interest to individuals, as well as medical professionals and researchers. The paper is well elaborated. The introduction is detailed enough to get into the topic quickly. The research is well presented and the results are presented in a way that is easy to follow.

This manuscript is a resubmission of an earlier submission. The following is a list of the peer review reports and author responses from that submission.

Round 1

Reviewer 1 Report

The contribution by Turek reflects upon some important aspects of manufacturing that practitioners deal with while preparing anatomical pieces. However, prior to publication, a number of issues need to be addressed:

  1. In Figure 1 the description of different anatomical parts requires much more detail.
  2. The details of the laser scanning system are rather scarce and also how the two pieces in Figure 2 assemble together.
  3. Repeatability is indicated as an important parameter, however, it is not well described in the materials and methods section.
  4. The details of the anatomical models that are no fully described in the text need to be added as an appendix
  5. Errors need to be properly defined with equations in the materials and methods section.
  6. Figures 6 to 10 are really hard to grasp at once, the author needs to explore other types of plots to make them more understandable.
  7. The discussion is not comprehensive enough to really capture the value of the precision manufacturing introduced here. Therefore, a more detailed comparison with the literature is required.
  8. The conclusions are too general and purely qualitative.

Reviewer 2 Report

Unusual for all of this type of analysis to have been performed by a single author - has everyone been credited?

English language issues in the text that need to be addressed.

Introduction seems somewhat broad - should be focused on the approach presented. There is insufficient details on the background of the production methods and application of the paper.

Why are these designs needed? Why are these methods selected? Why these production techniques?

The motivation for this approach is very unclear. Insufficient explanation is provided.

The results presented are difficult to follow, the methods are poorly presented and the data is generally insufficient in detail and quality for a journal paper.

Discussion - limited in scope and difficult to follow.

Conclusions - insufficient in detail and limited in content.